# Lactose Intolerance Assessed by Analysis of Genetic Polymorphism, Breath Test and Symptoms in Patients with Inflammatory Bowel Disease

**DOI:** 10.3390/nu13041290

**Published:** 2021-04-14

**Authors:** Olga Maria Nardone, Francesco Manfellotto, Caterina D’Onofrio, Alba Rocco, Giovanni Annona, Francesca Sasso, Pasquale De Luca, Nicola Imperatore, Anna Testa, Roberto de Sire, Elio Biffali, Fabiana Castiglione

**Affiliations:** 1Gastroenterology, Department of Clinical Medicine and Surgery, University Federico II of Naples, 80131 Naples, Italy; albertavittoria@libero.it (C.D.); a.rocco@unina.it (A.R.); francesca.sasso@libero.it (F.S.); annatesta82@virgilio.it (A.T.); roberto.desire@libero.it (R.d.S.); fabcasti@unina.it (F.C.); 2Sequencing and Molecular Analyses Center, RIMAR Department, Stazione Zoologica A. Dohrn, Villa Comunale, 80122 Naples, Italy; francesco.manfellotto@szn.it (F.M.); giovanni.annona@szn.it (G.A.); pasquale.deluca@szn.it (P.D.L.); elio.biffali@szn.it (E.B.); 3Gastroenterology and Endoscopy Unit, AORN Antonio Cardarelli, 80131 Naples, Italy; nicola.imperatore@alice.it

**Keywords:** hydrogen breath test, lactose intolerance, inflammatory bowel disease, lactase polymorphism

## Abstract

Many patients with inflammatory bowel disease (IBD) restrict dairy products to control their symptoms. The aim of the study was to investigate the prevalence of lactose intolerance assessed with hydrogen breath test (H-BT) in IBD patients in clinical remission compared to a sex, age and BMI matched control population. We further detected the prevalence of three single nucleotide polymorphisms of the lactase (LCT) gene: the lactase non persistence LCT-13910 CC (wildtype) and the intermediate phenotype LCT-22018 CT and LCT-13910 AG; finally, we assess the correlation between genotype and H-BT. A total of 54 IBD patients and 69 control who underwent clinical evaluation, H-BT and genetic test were enrolled. H-BT was positive in 64.8% IBD patients and 62.3% control (*p* = 0.3). The wild-type genotype was found in 85.2% IBD patients while CT-22018, AG-13910 and CT-22018/AG-13910 polymorphisms were found in 9.3%, 1.8% and 3.7%. In the control group, the wild-type genotype, CT-22018, AG-13910 and CT-22018/AG-13910 polymorphisms were found in 87%, 5.8%, 5.8% and 1.4% of cases, respectively. Therefore, the wild-type and polymorphisms’ prevalence did not differ between IBD population and control group (85.2% vs. 87%, *p* = 0.1) (14.8% vs. 13%, *p* = 0.7). The correlation between positive H-BT and genetic analysis showed that the wild-type genotype was associated with higher rate of lactose intolerance in the total population (OR 5.31, 95%CI 1.73–16.29, *p* = 0.003) and in the IBD (OR 7.61, 95%CI 1.36–42.7, *p* = 0.02). The prevalence of lactose intolerance in IBD patients did not differ from that of control. Despite suggestive symptoms, about 1/3 of IBD patients are not lactose intolerant, thus not needing “a priori” elimination diet. This may encourage a rationale and balanced dietary management in IBD.

## 1. Introduction

An overall prevalence of 35% patients with inflammatory bowel disease (IBD), despite in remission, experience gut symptoms like abdominal pain, bloating, diarrhea [1]. The etiology of these gastrointestinal (GI) symptoms in patients with quiescent IBD can be multifactorial, and it is challenging to differentiate if they are due to food intolerances, malabsorption and/or concomitant irritable bowel syndrome (IBS) [2].

Several studies have shown that about one-third of patients with ulcerative colitis (UC) and a half of those with Crohn’s disease (CD) in remission met the Rome III criteria for diagnosis of IBS at any one time point [3].

Effective management strategies for these IBD patients overlapping IBS include accurate exclusion of food intolerance. The most common food intolerance is the lactose intolerance. Indeed, it has been reported a prevalence worldwide at 68% with wide variation between different regions and an overall frequency of around two-thirds of the world’s population [4].

Lactase is an enzyme, expressed on the brush border of villi, able to hydrolyze the lactose into galactose and glucose, subsequently they are both absorbed in the small intestine. When lactose is not digested, it can be fermented by gut microbiota causing osmotic diarrhea and determining the production of gasses including hydrogen (H2), carbon dioxide (CO2), methane (CH3) and short chain fatty acids (SCFA) that leads symptoms like abdominal pain, bloating, flatulence [5]. Classically lactose intolerance is defined as the occurrence of typical intestinal symptoms such as abdominal pain, bloating and diarrhea after a lactose challenge in individuals with lactose malabsorption [5]. To date, the most reliable test to diagnose lactose malabsorption is the hydrogen breath test (HBT) [6]. In addition, the genetic test, that analyzed several single nucleotide polymorphisms (SNPs) upstream the lactase (LCT) gene, may be also used. Indeed, lactase persistence and non-persistence depend on the presence of genetic polymorphisms: lactase non-persistent adults are homozygous for an autosomal recessive allele that causes the post-weaning decline of lactase activity, while lactase persistent subjects are either hetero- or homozygous for a dominant allele that allows lactase to persist [7].

Understanding the prevalence of lactose intolerance in IBD population may result in important implications by preventing unnecessary restrictive diets and thereby contributing to the prevention of future complications, including malnutrition and the occurrence of calcium phosphate metabolism disorders. Accordingly, the aim of the study was to investigate the prevalence of lactose intolerance in IBD patients in clinical remission with symptoms suggestive for lactose intolerance, in comparison to a control population with the same symptoms. Furthermore, we analyzed three single nucleotide polymorphisms of the LCT gene, the lactase non persistence LCT-13910 CC (genetic wildtype) and the intermediate phenotype LCT-22018 AG and LCT-13910 CT, to determine their prevalence in IBD population and assess the correlation between genotype and hydrogen breath test.

## 2. Materials and Methods

### 2.1. Study Design

We conducted an observational, prospective study enrolling all consecutive IBD patients referred between 2018–2019 to an academic tertiary IBD center at the University Federico II of Naples. Patients aged ≥18 years with a histologically confirmed diagnosis of IBD were eligible. All of them were in remission even though they experienced the following gut symptoms: bloating, abdominal pain and diarrhea. Remission state was defined as Crohn’s disease activity Index (CDAI) < 150 for CD [8] and a partial Mayo score ≤ 1 for UC [9,10]. IBD population was compared with a sex- age- and BMI-matched control subjects, consisting of consecutive patients referred to our outpatient clinic for intestinal symptoms suggestive of lactose intolerance.

Of note, patients with significant comorbidities, malignancy, abdominal surgeries, pregnant or lactating were excluded. In addition, IBD active patients were excluded because of lactose intolerance secondary to a damage of the small intestinal mucosa.

Demographic data, including age, body mass index (BMI), smoke habits, disease characteristics (extent of disease, Montreal classification), treatments (corticosteroids, immunosuppressive therapy, biologics) and clinical disease activity scores (CDAI and partial Mayo score) were collected. At the time of the study’s inclusion, IBD patients and control usually consume milk and both fermented and non-fermented dairy products.

All patients underwent HBT. Of note, antibiotics, laxatives, proton pump inhibitor and probiotics were not allowed two weeks before HBT. The test was conducted in the morning, after 8 to 12 h fasting without performing physical activity and smoking. Before starting the test, patients washed their mouths with 20 mL of 0.05% chlorhexidine. A total of 9 measurements were carried out every 30 min for 4 h maximum after oral ingestion of 25 g of lactose.

A Quintron Model Breath Tracker DP Microlyzer gas chromatograph (Quintron Instruments, Milwaukee, WI, USA) was used to measure in parts per million (ppm) the H2 and CH4 concentration in breath samples. Lactose malabsorption was diagnosed when H2 increased > 20 ppm or CH4 > 10 ppm over baseline values [11]. A baseline H2 value > 10 ppm was defined as an exclusion criterion. In addition, a self-administered questionnaire based on a dichomotic scale “yes” or “no” to the question “Do you have abdominal discomfort such as abdominal pain, bloating, diarrhea after intake of milk or dairy products?” [12] was used for symptom assessment during the HBT.

Accordingly, the presence of GI symptoms after lactose ingestion associated with a positive HBT was defined lactose intolerance.

To detect genetic polymorphism, serum samples were obtained from each IBD patients and control and the SNaPshot^®^ Multiplex System, a primer extension-based method was used.

We analyzed eight different SNPs (single nucleotide polymorphisms), in some cases adjacent each other, in the genomic sequence, which are associated with lactase persistence [13,14,15] in Caucasian, Arabian Bedouins, sub-Saharian Africans and Asian populations, and then, we selected the three most relevant SNPs for the subsequent analysis.

Two primer pairs were designed to amplify the two regions, respectively of 400 bp and 250 bp, containing the SNiPs of interest in the lactase enhancer:(1)eLac12F—ACTACTCCCCTTTTACCTCGTT,eLac12R—TCTGTTTATCTCTGCTCTCATCAT,amplifying the 400 bp region containing the −13910 position (rs4988235);(2)eLac22F—AGCTGGGACCACAAGCAC,eLac21R—CATTATCAGCCAACATCAAAGCamplifying the 250 bp region containing the −22018 position (rs182549).

PCR protocol: initial step of denaturation at 95 °C for 3′, then 35 cycles of denaturation at 95 °C for 30″, annealing at 57 °C for 30″, elongation at 72 °C for 45″, followed by a final elongation at 72 °C for 5′ in a final volume of 20 µL with 0.5 µM of each primer, 1 unit of Taq Expand High Fidelity PCR System (Roche) and 0.5% DMSO. For each sample, 5 µL of each PCR product were mixed and purified with ExoSap (Thermo Fisher, Waltham, MA, USA), following manufacturer’s protocol.

The following primers were designed for SNP detection:


**Primer Name**

**SNP**

**Expected Variation**

**Sequence**

**Final Length**
eLac1rs41525747G/CAGGAGAGTTCCTTTGAGGCCA36eLac2rs4988236G/AGGAGAGTTCCTTTGAGGCCAG41eLac3rs4988235G/AGAGTTCCTTTGAGGCCAGGG46eLac4rs41456145A/GCCTTTGAGGCCAGGGGCT51eLac5rs773131166C/TCCTTTGAGGCCAGGGGCTA56eLac6rs41380347A/CCTTTGAGGCCAGGGGCTAC61eLac7rs145946881C/GGGTATTAAATGGTAACTTACGTCTTTATG66eLac8rs182549C/TACAAAGGTGTGAGCCACCG71

To obtain the desired length, a GACT repeat was added to each primer. The reactions were performed using the SNaPshot Multiplex Kit (Thermo Fisher, Waltham, MA, USA) following manufacturer’s protocol, with a final concentration of 0.2 µM for each primer (0.4 µM for the primers eLac4 and eLac5). Then the samples were treated with Calf Intestinal Alkaline Phosphatase (Thermo Fisher, Waltham, MA, USA). In 2 µL of each sample were added 0.2 µL of GeneScan™ 120 LIZ™ dye Size Standard (Applied Biosystems, Foster City, CA) and run on the 3730 DNA Analyzer (Thermo Fisher, Waltham, MA, USA) following manufacturer’s instructions. Raw data were analyzed with the GeneMapper 5 software (Thermo Fisher, Waltham, MA, USA). All the data obtained were confirmed by Sanger sequencing both strands with BigDye Terminator v3.1 Cycle Sequencing Kit (Applied Biosystems, Foster City, CA, USA), using the same primers described previously, on the 3730 DNA Analyzer (Thermo Fisher, Waltham, MA, USA).

### 2.2. Ethical Considerations

All subjects gave verbal and written consent before participation and the proposal was approved by the local Ethics Committees (209/17).

### 2.3. Statistical Analysis

Statistical analysis was performed using the Statistical Package for Social Sciences (SPSS software v.15.0, Chicago, IL, United States) for Windows. The descriptive statistics included the determining of mean values and standard deviation (SD) of the continuous variables, as well as percentages and proportions of the categorical variables. Statistical significance was assessed using chi-squared to evaluate the differences between percentages or proportions statistics and ANOVA to evaluate the differences between means. Furthermore, the nonparametric Mann–Whitney and Wilcoxon tests were used to evaluate the differences for dichotomous and continuous variables. Finally, these statistical analyses were performed again splitting patients affected by CD and UC to detect differences between each other. All the differences were considered significant when *p* ≤ 0.05.

## 3. Results

### 3.1. Patient Demographics

The total number of patients recruited was 54 IBD (22 with CD and 32 with UC, 37% males with median age of 37.3 ± 14.7 years and a BMI of 24.07 ± 4.04). All of them were in clinical remission according to CDAI < 150 for CD and a partial Mayo score < 2 for UC. Twelve UC patients had pancolitis (E3) colitis, 11 proctosigmoiditis (E1) while 9 had left-side colitis (E2). With regards to CD, the majority (16) had ileal disease (L1), 5 ileo-colonic disease (L3) and 1 colonic disease (L2). No patients underwent abdominal surgery. Most (77.8%) IBD patients were treated with mesalamine, 12.9% with biologics, 5.6% with steroids and 5.6% with immunosuppressant, while a total number of 69 matched subjects (37.7% males with a mean age of 37.8 ± 15.2 years and a BMI of 23.5 ± 4.01) were enrolled as control. Demographic details are shown in Table 1.

### 3.2. Lactose Breath Test

H-BT was positive in 35 IBD patients (64.8%) and 43 control (62.3%) (*p* = 0.3). In addition, positive H-BT did not differ significantly in CD and UC patients (*p* = 0.8). During H-BT test, we further collected the occurrence of symptoms after lactose intake. Among IBD patients with positive H-BT, 25 (72%) experienced GI symptoms and thus were diagnosed as lactose intolerant, while 10 (28%) did not record any symptoms during the test. With regards to control group, 29 (68%) individuals with H-BT experienced symptoms, whereas 14 (32%) had no symptoms despite positive H-BT (Table 2).

A further analysis showed that in patients with positive H-BT, there was not significant difference between IBD and control group in hydrogen and/or methane raised (*p* = 0.2). In addition, the production of hydrogen and methane did not differ between CD and UC (*p* = 0.3).

### 3.3. Genetic Test

The genetic analysis revealed that 46 IBD patients (85.2%) had wild-type genotype while CT-22018, AG-13910 and CT-22018/AG-13910 polymorphisms were found in 9.3%, 1.8% and 3.7% of cases. In the control group, the wild-type genotype, CT-22018, AG-13910 and CT-22018/AG-13910 polymorphisms were found in 87%, 5.8%, 5.8% and 1.4% of the cases, respectively. Of note, there was no significant difference between IBD population and control group in terms of wild-type genotype (85.2% vs. 87%, *p* = 0.1) and polymorphisms’ prevalence (14.8% vs. 13%, *p* = 0.7) (Figure 1).

A further analysis showed that there was no difference in terms of polymorphism also between CD and UC patients. In detail, the wild-type genotype was found in 90.9% CD and 81.3% UC patients (*p* = 0.8). In addition, 9.1% of CD patients and 9.4% of UC had CT-22018 (*p* = 0.1) (Figure 1). Finally, we found that 3.1% of UC patients carried the genotype AG-13910, 6.2% CT-22018/AG-13910, while no patients with CD carried neither AG-13910 (*p* = 0.4) nor CT-22018/AG-13910 (*p* = 0.2) (Figure 2).

### 3.4. Correlation between Lactose Breath Test and Genotypes

In the overall study population, the wild-type genotype was associated with higher rate of lactose intolerance assessed by using H-BT (Odds Ratio OR 5.31, 95% CI 1.73–16.29, *p* = 0.003). This was confirmed also in the IBD population (OR 7.61, 95% CI 1.36–42.7, *p* = 0.02), but not in the control group (OR 4, 95% CI 0.90–17.68, *p* = 0.07).

Likewise, the presence of polymorphisms CT-22018, AG-13910 and CT-22018/AG-13910 was associated with lactose tolerance in the total population (*p* = 0.004) and in the IBD population (*p* = 0.03). Conversely in the control group, the presence of polymorphisms was not associated with lactose tolerance (*p* = 0.1).

### 3.5. Correlation between Symptoms and Breath Test and Genetic Test

No specific symptom (abdominal pain, bloating, diarrhea) experienced at baseline was associated with higher rate of positive H-BT in overall population (*p* = 0.9). This was also confirmed in IBD patients (*p* = 0.8) and control (*p* = 0.7). Similarly, no specific baseline symptom was associated with higher rate of wild-type genotype in overall population (*p* = 0.7) and both IBD patients (*p* = 0.8) and control (*p* = 0.6).

## 4. Discussion

The incidence of lactose intolerance in patients with IBD has been already investigated, but the obtained results remain controversial [16,17,18]. A meta-analysis by Szilagyi et al. [16] has shown an increased risk of lactose intolerance only in CD with small bowel involvement, whereas in UC patients the incidence does not differ from the control. Recently, a study involving children affected by IBD reported that the incidence of lactose intolerance does not differ in respect to the general population [19]. We found a similar prevalence in lactose intolerance in IBD patients as compared with control. Furthermore, no significant difference was observed between CD and UC. However, it is important to underline that we did not include patients who underwent IBD related surgeries because the bowel damage could lead to malabsorption and thereby secondary lactose intolerance [20].

Of note, the clinical relevance of genetic test for the diagnosis of lactose intolerance is a matter of debate [21,22,23]. Given that the lactase activity declines progressively over time, the genotype analysis is not enough to diagnose lactase intolerance [24]. Therefore, genetic analysis could be considered as a snapshot of a moment wherein the rate of lactase decline not necessarily has already reached the threshold for symptoms that may appear later in life [5]. However, several studies have shown a very good agreement between the genetic test and the lactose breath test [17,21,22,24].

In the current study, the distribution of genotypes did not differ significantly between IBD patients and control. The presence of wild-type genotype was associated with a higher rate of lactose intolerance assessed with H-BT in the total population and IBD patients while the genetic polymorphisms CT-22018, AG-13910 and CT-22018/AG13910 were associated with lactose tolerance in the total population and IBD population.

The mechanism leading to symptoms after lactose ingestion is complex as it involves the production of metabolites by bacteria. Interestingly, differences in bacterial metabolism influence the probability of developing symptoms such as diarrhea, bloating, abdominal pain as a result of lactose ingestion. Among the hundreds of colonic bacteria, *Bifidobacteria*, *Lactobacilli* and *Escherichia coli* have beneficial effect on the ability to metabolize lactose [5]. Of note, these bacterial fermentation results in production of SCFA that are implicated in immune regulation, glucose and lipid homeostasis and colonocyte differentiation. [5] However, since methane producing Archeobacteria play a key role in the IBD gut microbiota [17,25,26], we also investigate the rise of hydrogen and methane in IBD patients and control. Previously, Eadala et al. showed that IBD patients had a higher methane raised than hydrogen breath [17]. Nevertheless, we did not find any significant difference in raised hydrogen and methane supporting the hypothesis that the gut microbiota of IBD patients in remission did not differ from control. Furthermore, a possible explanation is related to the use of a more stringent cut-off of >10 ppm methane, in accordance with the recent literature [11], whereas the previous study selected a cut-off >5 ppm over the nadir.

It is noteworthy that in a considerable proportion of IBD patients (42%) and even control (34%), we reported the occurrence of symptoms even with negative H-BT. These highlight a “nocebo effect” based on the subjective suggestion that the ingestion of lactose could induce abdominal symptoms in those patients [27]. This idea is also enhanced by some social media dealing with symptoms potentially triggered by lactose. Thus, many patients do not actually have any real lactose intolerance; rather, they are self-diagnosed lactose intolerant. These misconceptions lead many people to unnecessarily stop their consumption of dairy products. We observed a similar rate of lactose intolerance in our IBD population compared to control; accordingly, we strongly believe that IBD patients do not need to exclude dairy products “a priori” form their diet without evidence of lactose intolerance. Indeed, in the last years many IBD patients, and also their physicians, have benefitted from diet as adjuvant treatment to control symptoms [28,29]. Many patients believe that lactose-containing foods may lead to abdominal symptoms, and thereby, they restrict dairy products believing they cause their symptoms. It has been reported that 76% of patients with IBD restrict food groups on their own, based on subjective intolerance and worsening of their disease [30]. Interestingly, several studies have investigated the impact of fermented and no fermented products on H-BT and symptoms. In most of them, breath-hydrogen production was significantly lower with fresh yoghurt compared to milk and pasteurized products [31,32]. In addition, fewer symptoms were observed after consumption of fresh yoghurt than milk [27,32]. This will result in a huge expansion of fermented dairy products and lactose-free dairy market, expected to reach a 9 billion turnover by 2022 [33]. Therefore, current recommendations and evidence are required to promote a rational dietary approach for management of IBD instead of self-restriction habits.

The main strength of the current study is the full assessment of lactose intolerance involving genetic test, measurement of breath hydrogen and collection of symptoms. In clinical practice, we often deal with IBD patients although in remission who describe a stereotyped GI symptoms pattern after the ingestion, even in small amounts, of milk or milk-derivative products. Thus, we sustain that before making restrictions, an objective assessment of lactose intolerance is required.

A limitation of the study is a relatively small number of IBD patients and control. However, we aimed to estimate the rate of lactose intolerance in a selected population of IBD patients in clinical remission and control with GI symptoms suggestive of lactose intolerance. Assessing the prevalence of lactose intolerance in the general population was beyond the scope of this exploratory study. Thus, we seek to provide a rational approach to screen lactose intolerance in IBD patients in clinical remission before eliminating milk and dairy products and thereby prevent important nutrient deficiencies. In addition, patients did not complete a validated questionnaire about symptoms associated with the consumption of dairy products at home (HQ), including fermented and non-fermented products. We further did not use a visual analogue scale with the global symptom score to assess the severity of symptoms and fully excluded the potential confounding effect of concomitant conditions. We did not analyze a correlation between extension and location of IBD and an increased risk of lactose intolerance. Nevertheless, given that all enrolled IBD patients were in clinical remission, we believe that the location of disease should have no effect.

## 5. Conclusions

In conclusion, the use of a balanced nutritional approach in IBD patients in clinical remission with GI symptoms involves screening individuals for food intolerances and promoting a tailored food intake [34] in order to prevent the development of future complications, including malnutrition and the occurrence of calcium phosphate metabolism disorders.

## Figures and Tables

**Figure 1 nutrients-13-01290-f001:**
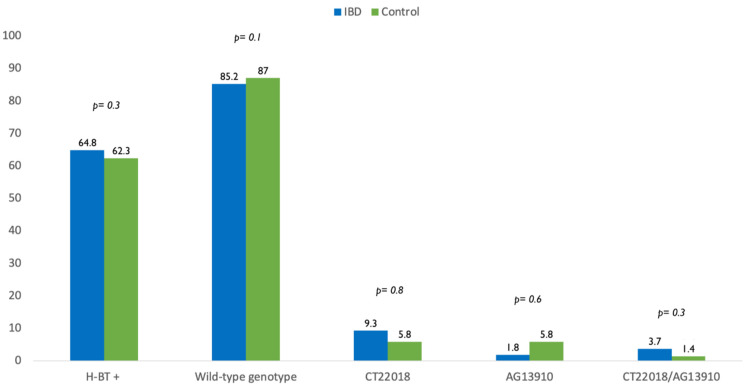
Difference in prevalence of positive breath test (H-BT +) and distribution of genotypes in inflammatory bowel disease (IBD) patients and control.

**Figure 2 nutrients-13-01290-f002:**
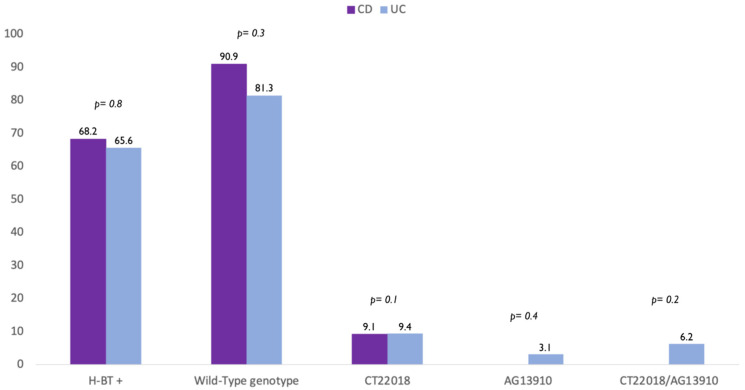
Difference in prevalence of positive breath test (H-BT +) and distribution of genotypes in patients with Crohn’s disease (CD) and Ulcerative colitis (UC) patients.

**Table 1 nutrients-13-01290-t001:** Baseline patient demographics.

		IBD * (n = 54)	Control Group (n = 69)	*p* Value
Males, n ^ξ^ (%)		20 (37%)	26 (37.7%)	0.9
Mean age, years ± SD ^β^		37.3 ± 14.7	37.8 ± 15.2	0.8
BMI ^¥^ (mean Kg/m^2^ ± SD)		24.07 ± 4.04	23.5 ± 4.01	0.4
Smoking habits	Yes, n (%)	9 (16.7%)	10 (14.5%)	0.8
	No/Ex, n (%)	45 (83.3%)	59 (85.5%)	0.1
Symptoms	Abdominal pain, n (%)	15 (27.8%)	19 (27.5%)	0.4
	Diarrhoea, n (%)	16 (29.6%)	20 (29%)	0.6
	Bloating, n (%)	23 (42.6%)	30 (43.5%)	0.3
Crohn’s disease, n (%)		22 (40.7%)		
Ulcerative colitis, n (%)		32 (59.3%)		
Montreal Classification	L1 °	16 (72.7%)		
	L2 °°	1 (4.5%)		
	L3 °°°	5 (22.7%)		
	L4 °°°°	0		
	E1 ^ץ^	11 (34.4%)		
	E2 ^ץץ^	9 (28.1%)		
	E3 ^ץץץ^	12 (37.5%)		
Medications	Mesalamine, n (%)	42 (77.8%)		
	Steroids, n (%)	3 (5.6%)		
	Immunosuppressants, n (%)	3 (5.6%)		
	Biologics, n (%)	7 (12.9%)		
	Antibiotics, n (%)	2 (3.7%)		
Positive H-BT ^¶^, n (%)		35 (64.8%)	43 (62.3%)	0.3
Genetics, n (%)	Wild type	46 (85.2%)	60 (87%)	0.1
	CT-22018	5 (9.3%)	4 (5.8%)	0.8
	AG-13910	1 (1.8%)	4 (5.8%)	0.3
	CT-22018/AG-13910	2 (3.7%)	1 (1.4%)	0.6

IBD *, inflammatory bowel disease; n ^ξ^, number; SD ^β^, standard deviation; BMI ^¥^ body mass index; L1 °, ileal location; L2 °°, colic location; L3 °°°, ileocolic location; L4 °°°°, ileocolic location; E1 ^ץ^, proctitis; E2 ^ץץ^, left-sided colitis; E3 ^ץץץ^, extensive colitis; H-BT ^¶^, hydrogen-breath test.

**Table 2 nutrients-13-01290-t002:** Symptoms recorded during the H-BT test in IBD patients and control group.

	IBD		Control	
	H-BT +	H-BT −	H-BT +	H-BT −
Symptoms	25 (72%)	8 (42%)	29 (68%)	9 (34%)
No symptoms	10 (28%)	11 (58%)	14 (32%)	17 (66%)

hydrogen breath test (H-BT); positive hydrogen breath test (H-BT +); negative hydrogen breath test (H-BT −); inflammatory bowel disease (IBD).

## Data Availability

The data presented in this study are available on request from the corresponding author.

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
