# Peer review of "Lactose Intolerance Assessed by Analysis of Genetic Polymorphism, Breath Test and Symptoms in Patients with Inflammatory Bowel Disease"

_nutrients, 2021, doi:10.3390/nu13041290_

Round 1
Reviewer 1 Report
The authors of this manuscript use approaches to assess lactose intolerance in patients with inflammatory bowel disease. Authors use a combination of hydrogen breath test and genetic polymorphism analysis to determine that there was no significant difference in the prevalence of lactose intolerance in IBD patients compared to the general public.
The authors conclude that exclusion of dietary lactose/dairy products is not likely to be beneficial for patients with IBD, based on the lack of differences between IBD and control patients in relation to lactose tolerance.
The sample size used in this study seems low to allow for any conclusions drawn to be meaningful. In this study the authors included a total of 35 IBD patients and 43 “control” patients. Based on the inability to control for diet, environment, or behavior in human cohorts, such a small sample size makes it difficult to draw reliable conclusions from the data. Additionally, the control group used in this study may not be the best control group use in order to answer the questions raised in this manuscript. The control group is discussed throughout the manuscript as “general population”, but these individuals are all symptom matched to IBD patients, which is not indicative of the general population. Based on the results, 62.3% of the general population controls, when data from NIH suggests that only approximately 36% of the general population have lactose malabsorption. This discrepancy skews the data significantly, suggesting that it is very likely that lactose intolerance in IBD patients is in fact much higher than that of the general population. These two issues should be included in the discussion.
There is no data in this manuscript that supports the notion that dietary restriction is detrimental to IBD patients. This fact must be made clear throughout the manuscript.
Finally there are instances of very unclear and confusing language, spelling errors, and gene nomenclature inconsistencies throughout this manuscript that make the manuscript difficult to read.
Minor:
Line 33: “1/3 of IBD patients is not lactose intolerant” should read “1/3 of IBD patients are not lactose intolerant.
Figure 1 and Figure 2: Legend states “breath test and polymorphism”, but at no point are these two features looked at together. If the authors do not want to present the prevalence of H-BT+ in non-WT individuals, I would re-word the legend to clarify that these two features are being conserved separately throughout the figures.
Line 209: a space is needed to separate polymorphism ID and the word “polymorphism”.
Lines 214-217: This statement is incredibly unclear and needs to be reworded. There is no description previously in the manuscript to what “OR” stands for.
Line 218: Nomenclature for polymorphisms seem to be flipped in this section compared to other sections throughout the manuscript. Authors need to use a consistent format in order to avoid confusion.
Line: 230: Reference is needed when discussing the controversy in lactose intolerance and IBD.
Line: 245: Authors state “test for lactase intolerance”. This should read “lactase persistence” or “lactose intolerance”.
Line 248-249: Heterozygosis when referring to CC and TT genotype intermediate (CT) may not be the correct terminology.
Line 260: This is a direct repetition of line 33, and needs to be changed accordingly.
Lines 263-264: This statement is poorly worded and confusing. Should be rewritten more clearly.
Line 274: Authors state “we did not include patients underwent IBD related surgeries”. Words seem to be missing from this sentence. “We did not include patients that underwent…” would be much clearer.
Lines 281-283: Authors discuss prevalence of osteoporosis in IBD patients, but do not include any comparison to non-IBD population. This comparison is critical to the discussion.
Lines 289-291: This entire paragraph needs to be rewritten. It is impossible to read in its current state.
Line 295: “We did not analysed” should read “we did not analyze”
Line 300: “score to asses” should read “score to assess”.
Reviewer 2 Report
General comment
This is a nice manuscript that, however, bases its results on a very limited number of study subjects. There are in my opinion some important problems with design, data evaluation and discussion. I doubt whether it adds much to existing literature on IBS and the “lactose problem”.
The main evaluation problem is that a positive hydrogen test shows lactose maldigestion, which is not a synonym of lactose intolerance (e.g. Szilagyi Nutrients 2015). It is known that lactose sensitivity is described equally in lactose digesters and maldigesters (Silanikove Nutrients. 2015).
More attention should have been paid to symptoms during the lactose tolerance test as the gold standard. This is acknowledged, since the authors say: “Accordingly, the presence of GI symptoms after lactose ingestion associated with a positive HBT was defined lactose intolerance”. It suggests that only the combination of a positive HBT and symptoms experienced during the test were taken for the diagnosis of lactose intolerance. This I can not derive from their results, which talk about positive HBT and symptoms during the test separately.
There is insufficient information on the control group. Where did these subjects come from? Consecutive patients, hospital, general practitioners, volunteers recruited how, etc. At present it is hard to be convinced that they represent the general population with the specified symptoms that equal those of IBD at first sight.
Detailed comment
English might occasionally be improved and there are some typos.
Figures 1 and 2: Prevalence of positive breath tests….
3.4 shows results for the correlation between the breath test outcomes and the various polymorphisms, but not lactase persistence per se (as such stated in the text). The polymorphisms are indeed related to lactase persistence (in the literature), but lactase persistence (enzyme activity in the gut) was formally not established in this study.
3.5 shows the correlation between symptoms at baseline and positive breath test. It would be informative to know also the relation with symptoms experienced during the test and a positive breath test (see comment above). Reason is that IBD patients may have higher sensitivity to e.g. pain. IBS often causes visceral sensitivity after lactose and nocebo effects. This misconception usually leads to an unnecessary reduction of dairy consumption (Corgneau Crit Rev Food Sci Nutr. 2017).
Paragraph beginning line 245. The Discussion on the decrease of lactase activity with age is somewhat strange given that all subjects were adults. The discussion might as well have been about bacteria that digest lactose in the gut. That would be more relevant, and so would be the usual consumption of dairy of the participants prior to testing. That is: fermented and non-fermented dairy
Paragraph beginning line 265. The symptoms and believes of the patients and their doctors might have been taken more seriously. Because science can not show any causes does not mean that there are none. We have so little knowledge: what if one finds in the future that it is not gas formation, but some other microbiota metabolite uniquely formed in IBD from lactose? If these patients in their minds improve because of that billion market, they by definition do.
Given the above, the present diagnosis of lactose intolerance and the many other limitations, the following statement, beginning line 272, is much too strong: “We showed that there is not an increased prevalence in lactose intolerance in IBD patients and thus they do not necessarily need to eliminate dairy products form their diet.”
Line 277: “Dairy foods may have several benefits in IBD: bone health, better control of hypertension, weight gain and a reduced risk for colorectal cancer either through calcium or vitamin D.” Cows milk contains little vitamin D, what is there is mostly added.
Line 279: “On the contrary insufficient vitamin D and a restricted intake of milk and dairy products may impact unfavourably on poor health outcomes like malnutrition and abnormal bone mineralization”. If we can not do without cows milk, one may wonder how most of the world thrives. Interestingly, only the countries with high intakes of milk, and thereby calcium, have osteoporosis. Prior to some 10,000 years before, cows milk was not consumed and ancient bones occasionally show low bone mass, and sometimes fractures, but no osteoporotic fragility fractures. The whole discussion on the beneficial effects of milk might be deleted, the more because there is also no discussion on fermented milk, like yoghurt, or lactobacilli, which were not tested by the authors either.
Also the comment on calcium might be deleted. The calcium RDA is much too high. A calcium RDA can not be established without taking into consideration other determinants like vitamin D, magnesium, acid/base, Na/K to mention a few.
Reviewer 3 Report
Manuscript "LACTOSE INTOLERANCE ASSESSED BY ANALYSIS OF GENETIC POLYMORPHISM, BREATH TEST AND SYMPTOMS IN PATIENTS WITH INFLAMMATORY BOWEL DISEASE" presents data on lactose intolerance in IBD patients that are clinically very important. The paper proves that despite microbiota changes in IBD and the fact that diseases affect mucosa of gut the lactose intolerance prevelance is the same in IBD and controls.
Minor revisions are needed: references starting from number 25 are presented in different style.
In the future it would be nice to confirm remission by endocsopy, clinical scoring systems are sometimes misleading.
Round 2
Reviewer 1 Report
The authors have mainly addressed the issues presented by this reviewer.
Reviewer 2 Report
I thank the authors for their comment and adjustments. Good luck with further studies.